# Policy optimization can be memory-efficient: LLM Alignment Through Successive Policy Re-weighting (SPR)

## Abstract

Reinforcement learning (RL) is serving as the cornerstone of aligning large language models (LLMs) to human behavior, by providing an appealing formulation and a suite of effective algorithms for learning behavior strategies through interacting with the underlying environment. Current paradigm of RL-based methods for LLM alignment, such as reinforcement learning with human feedback (RLHF) involves utilizing a reward function learned from extensive offline datasets to expedite the online training of reinforcement learning. The reward function learned is then used for policy optimization to obtain an improved policy (i.e. the LLM). Despite the success of RL approaches in aligning LLM with offline datasets, there are significant computational/limit of resources concern on applying RL-based methods for LLMs. For example, standard RLHF requires simultaneous loading of four models to the computing unit. In this paper, we develop a novel policy optimization algorithm named Successive Policy Re-weighting (SPR), matching the peak memory consumption of standard supervised fine-tune (SFT). Further, SPR can leverage both offline and online datasets to expedite online training and improve the sample efficiency. Specifically, SPR leverages a supervised learning subroutine to achieve policy improvement through re-weighting the policy according to the importance/performance of executed actions. Such simple and effective method is computationally inexpensive, requiring loading only one model at each update step, matching the computational cost of standard supervised fine-tuning procedure. Experimental results show that the proposed method can significantly outperform benchmark algorithms and accelerate the online training with available offline dataset.

## 1 Introduction

Aligning LLM to follow human instructions is of the central focus for artificial general intelligence (AGI). Recently, RL-based algorithms for LLM alignment, such as reinforcement learning with human feedback (RLHF, see Ouyang et al. (2022); Christiano et al. (2017)) are serving as the backbone of the recent success on the emergence of intelligence for LLMs such as GPT-4 (Achiam et al., 2023), Claude-3 (Anthropic, 2024) and Gemini-1.0 (Team et al., 2023). Among these methods, the most iconic RLHF utilizes large-scale preference datasets which consists of input prompt and two continuations, where one is preferred over the other, to train a reward model and use this reward model for policy

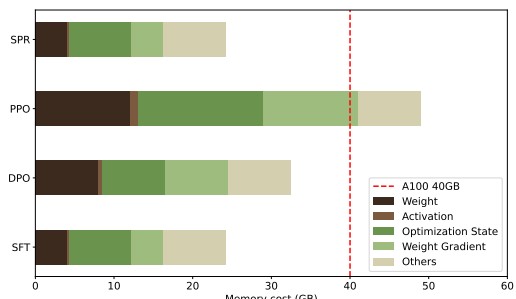

Figure 1: Estimated memory consumption of running the `Pythia-1B` model on `TL;DR` dataset with batch-size 2 on a single device, without checkpointing, memory offloading and distributed training such as Deepspeed. Here "SPR" is the implementation of our Algorithm 2.

optimization to iteratively improve LLMs. Therefore online policy optimization methods such as proximal policy optimization (PPO, see Schulman et al. (2017)) are the most prevalent in industry and the SOTA alignment approaches. On the other hand, Direct Preference Optimization (DPO,

see Rafailov et al. (2024)) simplifies RLHF by training the policy/LLM directly while implicitly learns the reward model via log of the ratio of likelihood between the learned model and a reference model. Both RLHF and DPO are successful in terms of improving the instruction-following and reasoning ability of LLMs (see, for example Ouyang et al. (2022); Tunstall et al. (2023); Dubey et al. (2024)) over the most straightforward supervised fine-tune (SFT) approach, which is analogous to the plain behavior cloning approach in RL literature (Pomerleau, 1988; Osa et al., 2018). LLMs aligned using RLHF are thus believed to open the current AI boom toward AGI (Bubeck et al., 2023).

There are several important distinctions between RL-based methods (such as RLHF) and DPO: First, RL-based methods require an explicit reward model for policy improvement, while DPO learns the policy directly; Second, RL-based method in general can effectively utilize an online pipeline using the explicit reward model, while the standard DPO is designed for offline data. Specifically for RL-based methods, a reward model could be trained by online and offline collected preference data using the standard Bradley-Terry model (Bradley & Terry, 1952). Given a reward model, RLHF utilizes standard policy optimization algorithm such as PPO at the policy optimization step, whose naive implementation requires loading four models at the same time, namely the reward, the policy, and two value function models. In contrast, DPO only requires loading two models (a policy and a reference model) simultaneously, but is arguably more sensitive to the distribution shift between the base model outputs and preference data, and it is not quite effective on challenging tasks (Xu et al., 2024; Lin et al., 2024; Ivison et al., 2024). Witnessing the success of RLHF, the memory efficiency issue of both RL-based method and DPO, as well as the limitation of offline nature of algorithms such as DPO, we pose the following question:

> **Can we reduce the compute and memory requirement of policy optimization to SFT level, while still leveraging an explicit reward function for effective online training?**

In this paper, we provide an affirmative answer to above question by proposing a novel algorithmic framework named **Successive Policy Re-weighting (SPR)**, which is a RL-based method, whose (peak) computational cost matches that of supervised fine-tuning (SFT). The proposed framework is developed from carefully examining the objective of TRPO/PPO and considers a constrained optimization reformulation from the original problem. A successive minimization algorithm is then proposed to iteratively optimize for the value function and the policy. When online data collection is available, we can collect online data from the current policy and add them to update the data buffer. This simple supervised learning update scheme makes off-policy online reinforcement learning more practical for real implementations. Specifically, our contributions are summarized as follows:

- We revisit the constrained policy optimization problem proposed in TRPO and a partial Lagrangian reformulation of such problem inspired by the advantage-weighted regression (AWR, see Peng et al. (2019)) method. We provide closed-form policy solution of the partial Lagrangian (see Theorem 1), and such a solution is exact and takes the normalization factor of the optimal policy into account (which is a departure from Peng et al. (2019), see Remark 2.2);
- We propose a novel iterative Successive Policy Re-weighting method (SPR, Algorithm 1). SPR utilizes an explicit reward to learn the Q-value function and the log-sum-exp value function, then learns the policy through re-weighting the policy probability based on the Q-value function. SPR only loads one (value or policy) model at every step, achieving policy optimization with the **same memory consumption as standard supervised fine-tuning (SFT)** (see Figure 1). We thus believe that the proposed SPR can serve as a powerful alternative to the memory-costly PPO for the RLHF pipeline when aligning LLMs.
- We conduct extensive numerical experiments to verify the effectiveness of the proposed method. Specifically, we compare the SPR with standard PPO and DPO, as well as a Best-of-N algorithm (Dong et al., 2023) on both 1b and 8b models. We compare both the reward value and win-rate of 1b model trained by different methods on a text-summarization dataset, and evaluate the 8b model trained by different methods on OpenLLM leaderboard. All our results show the superior performance of the proposed method over standard baselines.

## 2 METHODOLOGY

Our main objective is to efficiently conduct policy optimization with comparable level of memory consumption as SFT, while leveraging both offline dataset and online data generated by intermediate models to further fine-tune the policy. Towards this end, let us first review the state-of-the-art alignment methods[1] and the policy optimization problem formulations.

---

[1]Due to the page limit, we refer to Appendix A for a more comprehensive literature review.

## 2.1 PRELIMINARIES ON LLM ALIGNMENTS

We model the LLM as a policy $\pi$ in a Markov decision process (MDP). A MDP is defined by the tuple $(\mathcal{S}, \mathcal{A}, P, \mu, r, \gamma)$, which consists of the state space $\mathcal{S}$, the action space $\mathcal{A}$, the transition dynamics $P : \mathcal{S} \times \mathcal{A} \times \mathcal{S} \rightarrow [0, 1]$, the initial state distribution $\mu(\cdot)$, the reward function $r : \mathcal{S} \times \mathcal{A} \rightarrow \mathbb{R}$ and the discounted factor $\gamma \in (0, 1)$. Under a transition dynamics model $P$ and a policy $\pi : \mathcal{S} \rightarrow \Delta_{\mathcal{A}}$ where $\Delta_{\mathcal{A}}$ is the probability simplex on the action space, further define the corresponding state visitation measure as $d_\pi(s) := (1 - \gamma) \sum_{t=0}^{T} \gamma^t P^\pi(s_t = s | s_0 \sim \eta)$ for any state $s \in \mathcal{S}$. Here $T$ is horizon size, which can be any positive integer or $\infty$.

In the LLM context, $\mathcal{S}$ and $\mathcal{A}$ correspond to the space of input prompts and the space of output continuations, respectively. If we consider the entire input sentences as state space and the entire output continuations as action space, then there is no transition dynamics, i.e. horizon = 1. On the other hand, if we consider the token-level state/action space[2], i.e. the action is modeled as the next token in the sentence, the transition dynamics is deterministic and $P(s'|s, a)$ is always 1 if $s' = (s, a)$ and 0 otherwise. In all the experiments, we stick to the sentence-level state/action, where $T = 1$. The reward model $r$ is usually another LLM trained specifically to evaluate the score of given prompt/continuation tuples, and the policy model $\pi$ is the LLM to be optimized.

Now consider an LLM parameterized by $\theta$. Denote $\pi_\theta(a|s)$ as the probability of outputting $a$ given input prompt $s$, and we assume the horizon $T = 1$ for the rest of this section. The following discussions review three common procedures for fine-tuning LLM: (1) supervised fine-tuning (SFT) over demonstration dataset, (2) reinforcement learning with human feedback (RLHF) over preference dataset, and (3) direct preference optimization (DPO).

**SFT.** Given a *demonstration dataset* $\mathcal{D} := \{(s, a)\}$ collected from an expert policy $\pi^E$ (i.e. $a \sim \pi^E(\cdot|s)$), the SFT optimizes the following problem:

$$\max_\theta \ \ell_{\text{SFT}}(\theta) := \mathbb{E}_{(s,a)\sim\mathcal{D}} \left[ \log \pi_\theta(a|s) \right]. \tag{1}$$

The above problem shares the same optimal solutions with $\min_\theta \ \mathbb{E}_{s\sim\mu}[D_{\text{KL}}(\pi^E(\cdot|s) \| \pi_\theta(\cdot|s))]$ (see (17)), and the latter shows that SFT aims at imitating the demonstration dataset via minimizing the KL divergence. It is worth noting that the SFT stage described here is closely related to the imitation learning approach used in the RL literature for learning from demonstration (Osa et al., 2018), whose goal is to mimic the policy of an expert.

**RLHF.** Let $r_\phi(s, a)$ denote a reward model parameterized by $\phi$, which evaluates a given input and output pair $(s, a)$. Then the LLM can be fine-tuned by the following RL problem:

$$\max_\theta \ \ell_{\text{RL}}(\theta) := \mathbb{E}_{s\sim\mu, a\sim\pi_\theta(\cdot|s)} \left[ r_\phi(s, a) \right] - \beta \mathbb{E}_{s\sim\mu}[D_{\text{KL}}(\pi_\theta(\cdot|s) \| \pi_{\text{ref}}(\cdot|s))], \tag{2}$$

where $\pi_{\text{ref}}$ is a fixed reference model and $D_{\text{KL}}$ is the KL-divergence to regulate the policy around the reference policy $\pi_{\text{ref}}$. In practice, (2) is usually solved by standard policy optimization techniques such as PPO (Schulman et al., 2017).

To find an appropriate reward model $r_\phi(s, a)$, RLHF (see e.g., Christiano et al. (2017)) leverages a set of *preference dataset* $\mathcal{P} := \{(s, a_w, a_l)\}$, where each data contains a pair of output $a_w, a_l$, and $a_w$ is preferred over $a_l$ by human labeler (denoted as $a_w \succ a_l$). The Bradley-Terry model (Bradley & Terry, 1952) assumes that the probability of choosing $a_w$ over $a_l$ is

$$\mathbb{P}(a_w \succ a_l \mid s) = \frac{\exp(r(s, a_w))}{\exp(r(s, a_w)) + \exp(r(s, a_l))} = \sigma(r(s, a_w) - r(s, a_l))$$

where $\sigma$ is the Sigmoid function. One can formulate the following problem to find the reward model:

$$\max_\phi \ \ell_{\text{RM}}(\phi) := \mathbb{E}_{s\sim\mu, (a_l \prec a_w)\sim\pi^P(\cdot|s)} \left[ \log \left( \sigma\big(r_\phi(s, a_w) - r_\phi(s, a_l)\big) \right) \right]. \tag{3}$$

It is widely observed in the literature that, models trained via episodically learning the policy (2) and learning the reward (3) typically outperform those that are only trained using SFT (Ouyang et al., 2022). The reward model guides the performance of the LLM and allows a better generalization ability via the consistent input of the preference data from human labeler.

---

[2]Similar modeling can be considered if we model a dialogue as a sequence of state (questions) and actions (answers).

**DPO**. DPO (Rafailov et al., 2024) proposes to incorporate reward learning implicitly by utilizing the structure of the optimal solution of the RL problem (2). Specifically, (2) implies that the optimal policy should satisfy:

$$r(s,a) = \beta \log \left( \frac{\pi_\theta(a|s)}{\pi_{\text{ref}}(a|s)} \right) + \beta \log Z_\theta(s), \tag{4}$$

where $Z_\theta(s)$ is the partition function. Plugging this equation back to (3) we get the DPO loss:

$$\max_\theta \ \mathbb{E}_{s \sim \mu, (a_l \prec a_w) \sim \pi^P(\cdot|s)} \Big[ \log \Big( \sigma \big( \beta \log \Big( \frac{\pi_\theta(a_w|s)}{\pi_{\text{ref}}(a_w|s)} \Big) - \beta \log \Big( \frac{\pi_\theta(a_l|s)}{\pi_{\text{ref}}(a_l|s)} \Big) \big) \Big) \Big]. \tag{5}$$

**The difference between DPO and RLHF**. Let us have a brief discussion about the above DPO reformulation. First, it is important to note that the fact that optimal policy of (2) takes the form of (4) *implicitly* says that there *must exist* a reward model $r$ such that

$$\pi(a|s) = \frac{\pi_{\text{ref}}(a|s) \exp \left( \frac{1}{\beta} r_\phi(s,a) \right)}{\sum_{\tilde{a} \in \mathcal{A}} \pi_{\text{ref}}(\tilde{a}|s) \exp \left( \frac{1}{\beta} r_\phi(s,\tilde{a}) \right)}. \tag{6}$$

However, in the DPO formulation, after directly plugging (4) into (3), problem (5) *directly* optimizes the policy parameter $\theta$, it remains unclear whether the optimized policy continues to satisfy (6) during the entire optimization process. Therefore, a more rigorous way to recover the solution of (2) is to solve the following problem (by introducing the explicit optimal policy constraint):

$$\max_\phi \ \mathbb{E}_{s \sim \mu, (a_l \prec a_w) \sim \pi^P(\cdot|s)} \Big[ \log \Big( \sigma \big( \beta \log \Big( \frac{\pi(a_w|s)}{\pi_{\text{ref}}(a_w|s)} \Big) - \beta \log \Big( \frac{\pi(a_l|s)}{\pi_{\text{ref}}(a_l|s)} \Big) \big) \Big) \Big]$$

$$\text{s.t. } \pi := \arg\max_\pi \mathbb{E}_{s \sim \mu, a \sim \pi(\cdot|s)} [r_\phi(s,a)] - \beta \mathbb{E}_{s \sim \mu} [D_{\text{KL}}(\pi(\cdot|s) \| \pi_{\text{ref}}(\cdot|s))]. \tag{7}$$

The difference between (7) and (5) indicates that DPO in (5) is not exactly an RLHF scheme, but a supervised preference data-fitting scheme. In other words, the constrained optimization problem of RLHF is simplified and the solution space of DPO is actually larger than that of RLHF. Related discussion about limitation of DPO from different angles can also be found in Xu et al. (2024); Lin et al. (2024); Ivison et al. (2024).

## 2.2 Constraint Policy Optimization Formulation

We develop our problem formulation that substitutes the most memory-intensive RL step in (2) with the constrained optimization formulation in (14), which leads to the new policy optimization algorithm in the next section. Let us start from the most general case where the horizon $T$ is $\infty$, then the setting in our experiments ($T = 1$) will naturally follow. We inspect the original policy optimization by dropping the KL-divergence constraint in (2) and obtain:

$$J(\pi) := \mathbb{E}_{\tau \sim \pi} \Big[ \sum_{t=0}^{\infty} \gamma^t r(s_t, a_t) \Big], \tag{8}$$

where $\tau := (s_0, a_0, s_1, a_1, \cdots)$ denotes one trajectory, corresponding to one data point with prompt(s) and continuation(s). Under a policy/LLM $\pi$, we can define the corresponding value function $V^\pi$ and the Q-function $Q^\pi$ as below:

$$V^\pi(s) := \mathbb{E}_{\tau \sim \pi} \Big[ \sum_{t=0}^{\infty} \gamma^t r(s_t, a_t) \mid s_0 = s \Big], \tag{9a}$$

$$Q^\pi(s,a) := r(s,a) + \gamma \mathbb{E}_{s' \sim P(\cdot|s,a)} \big[ V^\pi(s') \big]. \tag{9b}$$

We can further define the advantage function for each state action pair $(s,a)$ as follows:

$$A^\pi(s,a) := Q^\pi(s,a) - V^\pi(s). \tag{10}$$

The fundamental idea of policy improvement is that, suppose there is a reference policy $\pi'$, we can maximize the performance gap over the reference policy to achieve policy improvement:

$$\eta_{\pi'}(\pi) := J(\pi) - J(\pi'). \tag{11}$$

It turns out that the performance improvement of the policy $\pi$ over the reference policy $\pi'$ can be expressed by the advantage function $A^{\pi'}(s,a)$.

**Lemma 1** (Lemma 1.16 in (Agarwal et al., 2019)). *For any policy $\pi$ and $\pi'$, the performance difference can be expressed as below:*

$$\eta_{\pi'}(\pi) = \frac{1}{1-\gamma}\mathbb{E}_{s\sim d_\pi(\cdot), a\sim\pi(\cdot|s)}\big[A^{\pi'}(s,a)\big] \tag{12}$$

*where $d_\pi(s) := (1-\gamma)\sum_{t=0}^\infty \gamma^t P^\pi(s_t = s | s_0 \sim \mu)$ denotes the state visitation measure.*

In the LLM context, this performance difference lemma indicates that to align the LLM with the reward model, i.e. to maximize the policy improvement objective in (11), one just need to seek for a policy $\pi$ which induces positive expected advantage $\mathbb{E}_{s\sim d_\pi(\cdot), a\sim\pi(\cdot|s)}\big[A^{\pi'}(s,a)\big] > 0$ over the reference policy $\pi'$. Therefore, we focus on maximizing (12).

However, from a practical point of view, the dependency on sampling data from the visitation measure $d_\pi(\cdot)$ makes it difficult to optimize the performance difference defined in (12). Following the trust region policy optimization (TRPO, see Schulman et al. (2015)) framework, we instead consider an approximation to $\eta_{\pi'}(\pi)$ by $\tilde{\eta}_{\pi'}(\pi)$:

$$\tilde{\eta}_{\pi'}(\pi) = \frac{1}{1-\gamma}\mathbb{E}_{s\sim d_{\pi'}(\cdot), a\sim\pi(\cdot|s)}\big[A^{\pi'}(s,a)\big] \tag{13}$$

where $d_{\pi'}(\cdot)$ denotes the state visitation measure under the reference policy $\pi'$. According to Theorem 1 in (Schulman et al., 2015), $\tilde{\eta}_{\pi'}(\pi)$ serves as a good approximation to $\eta_{\pi'}(\pi)$ when the two policies $\pi$ and $\pi'$ are close in terms of the KL-divergence. Thus, when maximizing the surrogate objective $\tilde{\eta}_{\pi'}(\pi)$ defined in (13) while penalizing the KL divergence between $\pi$ and $\pi'$, we are able to guarantee monotonic performance improvement at each policy iteration step.

Above discussions lead to the following optimization objective, which is also used in the TRPO: at each policy iteration step and given the previous policy $\pi_{\text{old}}$, one solves for the constrained policy optimization problem:

$$\max_\pi \quad \tilde{\eta}_{\pi_{\text{old}}}(\pi) \tag{14a}$$

$$s.t. \quad \mathbb{E}_{s\sim d_{\pi_{\text{old}}}(\cdot)}\big[D_{\text{KL}}\big(\pi(\cdot|s)||\pi_{\text{old}}(\cdot|s)\big)\big] \le \epsilon, \tag{14b}$$

$$\sum_{a\in\mathcal{A}} \pi(a|s) = 1, \forall s \in \mathcal{S}, \tag{14c}$$

$$\pi(a|s) \ge 0, \forall s \in \mathcal{S}, a \in \mathcal{A}. \tag{14d}$$

In the following theorem, whose proof is provided in the Appendix B, we show the closed-form expression of the optimal policy.

**Theorem 1.** *The optimal policy $\pi^*$ from (14a)-(14d) is*

$$\pi^*(a|s) = \pi_{\text{old}}(a|s)\exp\Big(\frac{1}{\beta}\big(Q^{\pi_{\text{old}}}(s,a) - W^{\pi_{\text{old}}}(s)\big)\Big) \tag{15}$$

*where $\beta := \frac{1}{(1-\gamma)\alpha}$ and $W^{\pi_{\text{old}}}(s) := \beta\log\Big(\mathbb{E}_{a\sim\pi_{\text{old}}(\cdot|s)}\Big[\exp\Big(\frac{1}{\beta}Q^{\pi_{\text{old}}}(s,a)\Big)\Big]\Big)$ is defined as a reference function dependent on the state $s$, also known as the log-sum-exp value function.*

*Remark 1.* Note that if we replace $W^{\pi_{\text{old}}}(s)$ by the value function $V^{\pi_{\text{old}}}(s) := \mathbb{E}_{a\sim\pi_{\text{old}}(\cdot|s)}Q^{\pi_{\text{old}}}(s,a)$, we arrive at Peng et al. (2019, equation (8)):

$$\pi^*(a|s) = \frac{1}{Z(s)}\pi_{\text{old}}(a|s)\exp\Big(\frac{1}{\beta}\big(Q^{\pi_{\text{old}}}(s,a) - V^{\pi_{\text{old}}}(s)\big)\Big)$$

where $Z(s)$ is the partition function to normalize the policy. $Z(s)$ is not negligible when estimating the optimal policy, which motivates us to use the log-sum-exp function $W^{\pi_{\text{old}}}(s)$ instead. Similar trick can be observed in maximum entropy RL literature such as Garg et al. (2022). Theorem 1 frees us from estimating the impractical partition function $Z(s)$ and opens the gate for efficient algorithm for solving for the optimal policy, by solving the Q and W functions in (15) respectively.

With the closed-form optimal policy in Theorem 1, the remaining question is how to develop a simple and efficient algorithm to approximate the optimal policy at each policy iteration. In the next section, we develop a practical algorithm to approximate the optimal policy $\pi^*$ defined in (15), which also optimizes the original objective (8).

## 3 ALGORITHM DESIGN

In this section, we design a memory-efficient algorithm to approximate the optimal policy $\pi^*$ defined in (15). Despite the fact that it has a closed-form solution, it remains unclear how to efficiently estimate the the Q and W functions on the right hand side of (15).

**Algorithm Derivation.** Now suppose we parameterize our policy by parameter $\theta$ in practice. Since the optimal policy for the constrained optimization problem defined in (14a)-(14d) enjoys a closed-form solution in (15), we directly approximate the optimal policy $\pi^*$ corresponding to a reference policy $\pi_{\text{old}}$ through solving the following KL divergence minimization problem:

$$\min_\theta \ \mathbb{E}_{s \sim d_{\pi_{\text{old}}}(\cdot)}\Big[D_{\text{KL}}\big(\pi^*(\cdot|s)||\pi_\theta(\cdot|s)\big)\Big]. \tag{16}$$

Based on the definition of the KL divergence $D_{\text{KL}}\big(\pi^*(\cdot|s)||\pi_\theta(\cdot|s)\big) = \mathbb{E}_{a \sim \pi^*(\cdot|s)}\big[\log \frac{\pi^*(a|s)}{\pi_\theta(a|s)}\big]$, we can obtain the following relations:

$$\mathbb{E}_{s \sim d_{\pi_{\text{old}}}(\cdot)}\Big[D_{\text{KL}}\big(\pi^*(\cdot|s)||\pi_\theta(\cdot|s)\big)\Big] = \mathbb{E}_{s \sim d_{\pi_{\text{old}}}(\cdot), a \sim \pi^*(\cdot|s)}\Big[\log \pi^*(a|s) - \log \pi_\theta(a|s)\Big]. \tag{17}$$

Eq. (17) implies that minimizing the KL divergence between $\pi^*$ and $\pi_\theta$ is equivalent to solving the following maximum likelihood estimation (MLE) problem:

$$\max_\theta \ L_{\text{MLE}}(\theta) := \mathbb{E}_{s \sim d_{\pi_{\text{old}}}(\cdot), a \sim \pi^*(\cdot|s)}\big[\log \pi_\theta(a|s)\big]. \tag{18}$$

This implies that solving (18) is equivalent to solving (16), and the resulting policy $\pi_\theta$ approximates $\pi^*$. However, one critical issue in solving the MLE problem is that we are not able to sample actions from the optimal policy $\pi^*$ even though the closed-form expression of $\pi^*$ is available in (15).

In order to resolve this issue, we leverage the technique from importance sampling where we sample actions from the existing policy $\pi_{\text{old}}$ and then weigh each sample $(s, a)$ by its importance weight $\frac{\pi^*(a|s)}{\pi_{\text{old}}(a|s)}$. Towards this end, we write:

$$L_{\text{MLE}}(\theta) = \mathbb{E}_{s \sim d_{\pi_{\text{old}}}(\cdot), a \sim \pi^*(\cdot|s)}\big[\log \pi_\theta(a|s)\big] = \mathbb{E}_{s \sim d_{\pi_{\text{old}}}(\cdot), a \sim \pi_{\text{old}}(\cdot|s)}\Big[\frac{\pi^*(a|s)}{\pi_{\text{old}}(a|s)} \log \pi_\theta(a|s)\Big]. \tag{19}$$

Leveraging the relation between $\pi^*$ and $\pi_{\text{old}}$ as shown in (15), $L_{\text{MLE}}(\cdot)$ can be rewritten as:

$$L_{\text{MLE}}(\theta) = \mathbb{E}_{s \sim d_{\pi_{\text{old}}}(\cdot), a \sim \pi_{\text{old}}(\cdot|s)}\Big[\frac{\pi^*(a|s)}{\pi_{\text{old}}(a|s)} \log \pi_\theta(a|s)\Big]$$

$$= \mathbb{E}_{s \sim d_{\pi_{\text{old}}}(\cdot), a \sim \pi_{\text{old}}(\cdot|s)}\Big[\exp\Big(\frac{1}{\beta}\big(Q^{\pi_{\text{old}}}(s, a) - W^{\pi_{\text{old}}}(s)\big)\Big) \log \pi_\theta(a|s)\Big]. \tag{20}$$

With the state-action pairs $(s, a)$ sampled from the policy $\pi_{\text{old}}$, we approximate the corresponding optimal policy through optimizing the objective in (20). In the LLM context, (20) can be viewed as a *re-weighting* process, where we increase or decrease the log likelihood of the previous policy $\pi_{\text{old}}$ according to the approximate advantage function $Q^{\pi_{\text{old}}} - W^{\pi_{\text{old}}}$, i.e. we increase the likelihood of actions/continuations with higher rewards than the baseline and decrease the ones with lower rewards.

In order to train a parameterized policy $\pi_\theta$ through optimizing the supervised learning objective in (20), it remains to estimate the Q-function $Q^{\pi_{\text{old}}}(s, a)$ and the reference function $W^{\pi_{\text{old}}}(s)$. To estimate the Q-function $Q^{\pi_{\text{old}}}(s, a)$ under a policy $\pi_{\text{old}}$, we solve the fixed point of the following Bellman operator $\mathcal{B}$:

$$\mathcal{B}^{\pi_{\text{old}}}Q(s, a) = r(s, a) + \gamma \mathbb{E}_{s' \sim P(\cdot|s,a), a' \sim \pi_{\text{old}}(\cdot|s')}\big[Q(s', a')\big]. \tag{21}$$

Following Fujimoto et al. (2018); Haarnoja et al. (2018), we use the temporal difference learning to minimize Bellman error to approximate $Q^{\pi_{\text{old}}}(s, a)$ by a parameterized Q-function $Q_\phi(s, a)$.

Next, let us estimate the reference function $W^{\pi_{\text{old}}}(s)$ under the policy $\pi_{\text{old}}$. Based on the definition of $W^{\pi_{\text{old}}}(s)$ in (15), a naive way is to construct empirical estimations to approximate the reference function. For each state $s$, one can sample a small batch of $N$ actions from policy $\pi_{\text{old}}(\cdot|s)$ and then construct the empirical estimate $\widehat{W}^{\pi_{\text{old}}}(s)$ defined as below:

$$\widehat{W}^{\pi_{\text{old}}}(s) := \beta \log \Big(\sum_{i=1}^N \exp\big(\frac{1}{\beta}Q^{\pi_{\text{old}}}(s, a_i)\big)\Big). \tag{22}$$

To obtain an accurate estimate of the reference function $W^{\pi_{\text{old}}}(s)$, it is necessary to make the batch-size $N$ large enough, which is computationally expensive. The estimation error between $\widehat{W}^{\pi_{\text{old}}}(s)$ and $W^{\pi_{\text{old}}}(s)$ are not negligible when the batch-size $N$ is small, see Nair et al. (2020).

The drawback of the empirical estimator in (22) motivates us to estimate the reference function $W^{\pi_{\text{old}}}(s)$ through parameterized approximations. Towards this end we have the following claim:

**Claim.** $W^{\pi_{\text{old}}}(s)$ satisfies

$$W^{\pi_{\text{old}}}(s) = \arg\min_w \ \mathbb{E}_{a \sim \pi_{\text{old}}(\cdot|s)}\Big[ \exp\Big(\frac{Q^{\pi_{\text{old}}}(s,a) - w}{\beta}\Big) - \frac{Q^{\pi_{\text{old}}}(s,a) - w}{\beta} - 1\Big]. \qquad (23)$$

**Derivation of the claim.** We observe that the reference function $W^{\pi_{\text{old}}}(s)$ shares the same expression to the fitted parameter of Gumbel distribution by maximum likelihood estimation. According to Coles et al. (2001); Forbes et al. (2011); Garg et al. (2022), when we model a random variable $x$ by Gumbel distribution $x = h - \epsilon$ where $\epsilon \sim \mathcal{G}(0, \beta)$ is a Gumbel noise, the log-likelihood has the following expression:

$$\mathbb{E}_x\Big[ \log p(x; h, \beta)\Big] = \mathbb{E}_x\Big[ \log\Big(\frac{1}{\beta}\exp\big(\frac{x-h}{\beta} - \exp(\frac{x-h}{\beta}))\big)\Big)\Big]. \qquad (24)$$

To fit the location parameter $h$ through maximum likelihood estimation, we can minimize the following loss function:

$$\min_h \ f(h) := \mathbb{E}_x\Big[ \exp\big(\frac{x-h}{\beta}\big) - \frac{x-h}{\beta} - 1\Big] \qquad (25)$$

where the loss function $f(h)$ shares similar form to the Linex loss in the literature of econometrics (Parsian & Kirmani, 2002; Chang & Hung, 2007). Through solving the loss function $f(h)$ defined in (25), we obtain the maximum likelihood estimator of the location parameter as $\hat{h} = \beta \log \mathbb{E}_x\big[ \exp\big(\frac{x}{\beta}\big)\big]$. Here, we observe that the estimate $\hat{h}$ shares the same expression to the reference function $W^{\pi_{\text{old}}}(s)$ defined in (15). This shows that for a fixed state $s$, the value of $W^{\pi_{\text{old}}}(s)$ solves the following optimization problem in (23).

Now given a parameterized Q-function $Q_\phi(s,a)$ which approximates the exact Q-function $Q^{\pi_{\text{old}}}(s,a)$, we can train a parameterized reference function $W_\varphi(s)$ to approximate $W^{\pi_{\text{old}}}(s)$ for all states $s \in \mathcal{S}$ through optimizing the following loss function:

$$\min_\varphi \ L(\varphi) := \mathbb{E}_{a \sim \pi_{\text{old}}(\cdot|s)}\Big[ \exp\Big(\frac{Q_\phi(s,a) - W_\varphi(s)}{\beta}\Big) - \frac{Q_\phi(s,a) - W_\varphi(s)}{\beta} - 1\Big]. \qquad (26)$$

This completes our derivation of the algorithm. Now, we are ready to summarize our proposed Successive Policy Re-weighting (SPR) algorithm in Alg. 1. We solve the value function $Q$ (parameterized by $\phi$) and reference function $W$ (parameterized by $\varphi$) by solving (21) and (26), respectively. Then we can solve for the policy/LLM $\pi_\theta$ by minimizing the re-weighting loss (20).

**Algorithm Simplification.** Note that if we model the entire input prompt and output continuation as the state and action respectively, the horizon will be automatically reduced to one. In this case, the $Q$ value function reduces to the reward function $r$ and line 4 of Algorithm 1 is discarded immediately, also $W$ in (22) becomes the log-sum-exp of the reward function $r$. The resulting *simplified* algorithm is presented in Algorithm 2, which can be implemented for LLM alignment. Notably, both Algorithm 1 and 2 only require a single model to be loaded in the memory for each of it update step.

**Discussion.** SPR significantly decreases the memory consumption of PPO by reducing the number of models loaded in the memory from four to one, matching the memory requirement of standard SFT. The proposed method saves more memory comparing to the DPO since DPO requires loading a froze reference model during the training step, while SPR only loads one model throughout its algorithm process; On the other hand, SPR still allows online data and explicit reward feedback through iterative sampling-and-fitting process, greatly increase its potential in alignment comparing to offline-data-only method such as SFT or DPO. The memory and time consumption of SPR comparing with related methods are summarized in Figure 1.

## 4 EXPERIMENTAL RESULTS

In this section, we demonstrate the effectiveness of our proposed method numerically.

---

**Algorithm 1** *Successive Policy Re-weighting (SPR)*

---

1: **Input:** Collect an demonstration dataset $\mathcal{D} = \{(s, a, s')\}$ and a given reward function $r(\cdot, \cdot)$.
2: Initialize the parameterized policy (LLM), Q-function and reference function as $\pi_\theta$, $Q_\phi$ and $W_\varphi$
3: **for** $k = 0, 1, \ldots, K - 1$ **do**
4: $\quad \phi_{k+1} = \arg\min_\phi \mathbb{E}_{(s,a,s')\sim\mathcal{D}, a'\sim\pi_{\theta_k}(\cdot|s')}\left[\left(Q_\phi(s,a) - r(s,a) - \gamma Q_{\bar\phi_k}(s',a')\right)^2\right]$
5: $\quad \varphi_{k+1} = \arg\min_\varphi \mathbb{E}_{s\sim\mathcal{D}, a\sim\pi_{\theta_k}(\cdot|s)}\left[\exp\left(\frac{Q_{\phi_{k+1}}(s,a) - W_\varphi(s)}{\beta}\right) - \frac{Q_{\phi_{k+1}}(s,a) - W_\varphi(s)}{\beta} - 1\right]$
6: $\quad \theta_{k+1} = \arg\min_\theta \mathbb{E}_{s\sim\mathcal{D}, a\sim\pi_{\theta_k}(\cdot|s)}\left[\exp\left(\frac{1}{\beta}\left(Q_{\phi_{k+1}}(s,a) - W_{\varphi_{k+1}}(s)\right)\right) \log \pi_\theta(a|s)\right]$
7: $\quad \bar\phi_{k+1} = \alpha\phi_{k+1} + (1 - \alpha)\bar\phi_k$
8: $\quad$ **if** $k >$ sampling threshold **then**
9: $\quad\quad$ sample trajectories $\tau_1, \cdots, \tau_N$ from the current policy $\pi_{\theta_{k+1}}$
10: $\quad\quad$ add trajectories to the data buffer: $\mathcal{D} \leftarrow \mathcal{D} \cup \{\tau_1, \cdots, \tau_N\}$
11: $\quad$ **end if**
12: **end for**

---

**Algorithm 2** *Successive Policy Re-weighting (SPR) for LLM alignment*

---

1: **Input:** Collect an demonstration dataset $\mathcal{D} = \{(s, a)\}$ and a given reward function $r(\cdot, \cdot)$.
2: Initialize the parameterized policy (LLM), Q-function and reference function as $\pi_\theta$, $Q_\phi$ and $W_\varphi$
3: **for** $k = 0, 1, \ldots, K - 1$ **do**
4: $\quad \varphi_{k+1} = \arg\min_\varphi \mathbb{E}_{s\sim\mathcal{D}, a\sim\pi_{\theta_k}(\cdot|s)}\left[\exp\left(\frac{r(s,a) - W_\varphi(s)}{\beta}\right) - \frac{r(s,a) - W_\varphi(s)}{\beta} - 1\right]$
5: $\quad \theta_{k+1} = \arg\min_\theta \mathbb{E}_{s\sim\mathcal{D}, a\sim\pi_{\theta_k}(\cdot|s)}\left[\exp\left(\frac{1}{\beta}\left(r(s,a) - W_{\varphi_{k+1}}(s)\right)\right) \log \pi_\theta(a|s)\right]$
6: $\quad$ **if** $k >$ sampling threshold **then**
7: $\quad\quad$ sample prompt-continuation pairs $(s_1, a_1), \cdots, (s_N, a_N)$ from the current policy $\pi_{\theta_{k+1}}$
8: $\quad\quad$ add prompt-continuation pairs to the data buffer: $\mathcal{D} \leftarrow \mathcal{D} \cup \{(s_1, a_1), \cdots, (s_N, a_N)\}$
9: $\quad$ **end if**
10: **end for**

---

### 4.1 EXPERIMENT SETUP

**Model and Datasets.** We conduct our experiment in two settings: First, we test with `EleutherAI/pythia-1b-deduped` model over `TL;DR` text-summarization dataset and evaluate using a reward model and the win rate over the reference summary; Second, we train with `LLaMA3-SFT` model over a mixture of various prompt datasets, including UltraFeedback Cui et al. (2023), HelpSteer Wang et al. (2023) and so on (see Dong et al. (2024) for the details of the training data). We test the performance of the 8b model trained by SPR over various downstream tasks.

**Evaluation.** For the first experiment setting, we use a model trained from `Pythia-6.9b` with `TL;DR` dataset[3] as the reward evaluator. Note that the experiment is conducted on 1b model and we believe that this 6.9b model is capable of serving as the ground truth for this text summarizing task. We also record the win rate of the model generated summary against the ground truth summary, evaluated by GPT-3.5-turbo-0125.

For the second experiment setting, we evaluated the aligned model trained from `LLaMA3-SFT` over the Open LLM Leaderboard (Beeching et al., 2023). Open LLM Leaderboard involves various downstream tasks to test the performance of LLM through different dimensions, where we (following Chen et al. (2024); Li et al. (2024)) evaluate an LLM based on six tasks: commonsense reasoning (Arc Clark et al. (2018), HellaSwag Zellers et al. (2019), Winogrande Sakaguchi et al. (2021)), multi-task language understanding (MMLU Hendrycks et al. (2020)), human falsehood mimic (TruthfulQA Lin et al. (2021)) and math problem solving (GSM8K, Cobbe et al. (2021)).

**Implementation detail.** Instead of training a reference function as in step 4 of Algorithm 2, we use the empirical estimate $\widehat{W}^{\pi_{\text{old}}}(s)$ in (22) by generating $N$ responses based on the current policy,

---

[3]see this link for the detail of how to obtain this reward model from the base model of `EleutherAI/pythia-6.9b-deduped`.

due to its efficiency in practice. In our experiment, considering the time efficiency and performance limitations due to the scaling law Kaplan et al. (2020), we split the whole training `TL;DR` dataset of evenly into 10 pieces, so 11.7K prompts are used to generate $N$ independent responses for training in each iteration, and iterate the training dataset for 3 epochs on each data pieces. For the 8b model, we used 20K prompts per iteration and trained for 2 epochs on each data pieces. When we exhaust all the data, we restart from the first split of the dataset[4]. We refer to Appendix C for more details.

## 4.2 RESULTS ON 1B MODELS

We present the performance of `EleutherAI/pythia-1b-deduped` model on four different algorithms, including SPR (Algorithm 2), Best-of-N (Dong et al. (2023)), DPO[5] and PPO[6]. We use the SFT pythia-1b model[7] as the initial model. Here Best-of-N refers to selecting the sample with highest score, as evaluated by the 6.9b reward model, to train the policy model, which serves as a strong baseline.

In Fig. 2, it can be seen that both SPR and Best-of-N algorithm outperform DPO in terms of reward score. However, while the Best-of-N algorithm struggles to consistently surpass DPO in win rate, SPR still demonstrates strong performance in win rate against reference summaries. Furthermore, as $N$ increases, both Best-of-N and SPR achieve better performance, attributed to a more accurate estimation of the reference function and improved sample quality. Although the reward score of all four algorithms continue to increase, the win rate essentially converges to a stationary level.

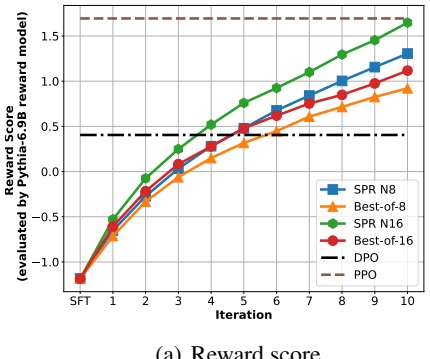
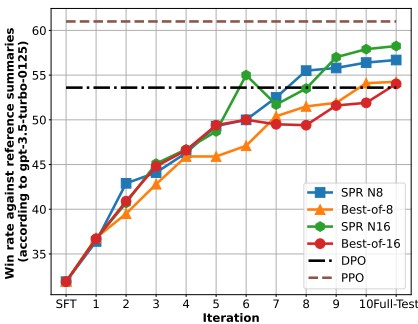

(a) Reward score

(b) Win rate against reference summaries

Figure 2: Left: The reward scores evaluated by Pythia-6.9b reward model trained on the `TL;DR` dataset on win rate of SPR algorithm on Pythia-1b model through the entire train split of the `TL;DR` dataset; Right: The win rate of our models' summaries over the human-generated reference summaries on the first 1000 test split of the `TL;DR` dataset, judged by GPT 3.5, "Full-Test" means evaluated on the whole test dataset. The x-axis represents the iteration number, each iteration uses 1/10 training data. To ensure fairness in testing, responses are generated using greedy search.

## 4.3 RESULTS ON 8B MODELS

In Figure 4 and Table 1, we compare the performance of our fine-tuned model by SPR with the base model `LLaMA3-SFT` on each task included in the leaderboard. SPR demonstrates superior performance compared to the Best-of-N approach and DPO after several iterations. Specifically, the average accuracy of SPR outperforms DPO by $0.22\%$ when $N = 16$ and by $0.77\%$ when $N = 32$, respectively.

Notably, our algorithm shows strong potential on the improving the LLM's ability of solving

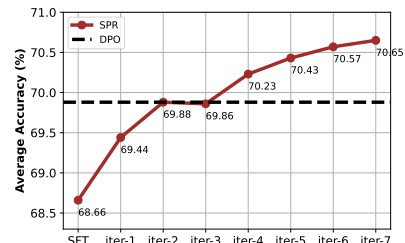

Figure 3: The average score of SPR with $N = 32$ and DPO baseline at different iterations on the Open LLM leaderboard datasets.

---

[4]For example, in Fig. 2, we exhaust all the training data when the "Iteration" on x-axis equals to 10.

[5]We use this pythia-1b DPO checkpoint.

[6]We use this pythia-1b PPO checkpoint.

[7]We use this pyhia-1b SFT checkpoint.

mathematical problem, with an almost $6\%$ improvement on the GSM8K task. In Figure 3, we show the average score of SPR algorithm with $N = 32$ compared to the DPO baseline over a total of 7 iterations. The results demonstrate a progressive increase in the average score, ranging from $68.66\%$ to $70.65\%$, indicating the effectiveness of our algorithm.

| Tasks
Metrics | Arc Challenge
acc_norm | TruthfulQA MC2
acc | Winogrande
acc | GSM8k
strict-match | HellaSwag
acc_norm | MMLU | Average |
|---|---|---|---|---|---|---|---|
| LLaMA3-SFT | 62.29% | 53.49% | **78.14%** | 72.55% | 81.03% | 64.49% | 68.66% |
| LLaMA3-DPO | **65.36%** | **60.02%** | 77.43% | 70.96% | 81.56% | 63.95% | 69.88% |
| Best-of-16 | 63.32% | 56.03% | 78.13% | 75.97% | 81.36% | 65.22% | 70.01% |
| SPR-16 | 63.57% | 56.14% | 77.74% | **76.12%** | **81.68%** | **65.34%** | **70.10%** |

Table 1: Performance of Policy in Open LLm Leaderboard for four different algorithms. SPR and Best-of-N algorithm with $N = 16$ are run over five iterations.

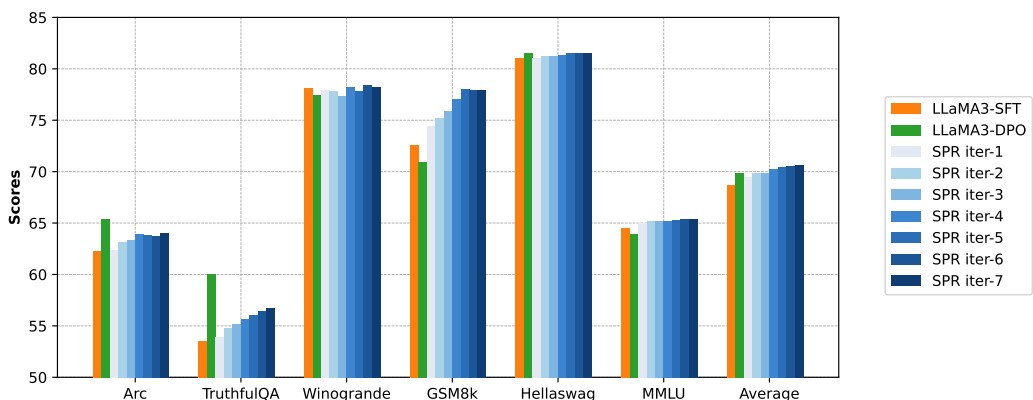

Figure 4: Performance comparison between DPO training and SPR with $N = 32$ across the six benchmark datasets.

## 5 CONCLUSIONS AND LIMITATIONS

In this paper we proposed a memory-efficient policy optimization approach named successive policy re-weighting (SPR), matching the peak memory consumption of standard supervised fine-tune (SFT). SPR is flexible for offline and online policy training and achieves state-of-the-art performance on our experiment for aligning 1b and 8b models. The computational time of the proposed method is in general higher than SFT or DPO, and the performance is largely dependent on the number of samples $N$ to estimate the reference function in (22). Future works include tuning SPR pipeline to make it work for very large LLMs (>50b) and exploring alignment methods that are both time- and memory-efficient.

## REPRODUCIBILITY STATEMENT

The base models and datasets in this paper are publicly available. However, our current implementation codes are experimental and not ready for releasing. We will release all our codes upon acceptance of this paper.

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

APPENDIX

# A    RELATED WORK

In this section, we discuss related work on both the LLMs and reinforcement learning with offline datasets.

**RL for LLMs alignment.** RL-based methods have show great success in improving the abilities of LLMs, including text-summarizing (Liu et al., 2020; Ziegler et al., 2019), story-telling (Ziegler et al., 2019), instruction-following (Ouyang et al., 2022; Ramamurthy et al., 2022), problem-solving (Trinh et al., 2024), etc. The most popular RL pipeline for LLM alignment is RLHF (Ouyang et al., 2022; Christiano et al., 2017), which models the reward using the popular Bradley-Terry model (Bradley & Terry, 1952) and optimize the policy by policy optimization methods, such as REINFORCE (Williams, 1992; Ahmadian et al., 2024; Li et al., 2023b), PPO (Schulman et al., 2017), GRPO (Shao et al., 2024) and R-max (Brafman & Tennenholtz, 2002). On the other hand, the extra computational cost introduced by RL-based method is not negligible. DPO (Rafailov et al., 2024) serves as a powerful substitute for RL-based method by directly optimizing for the LLM/policy (and the reward is implicitly represented by the policy), making the memory consumption as low as the standard SFT. However the absence of reward model in DPO pipeline is drawing discussions on its limited generalization ability in the LLM alignment community (Li et al., 2023a; Xu et al., 2024). Standard DPO utilizes an offline pre-collected preference dataset. Several other methods, such as KTO (Ethayarajh et al., 2024), sequence likelihood calibration (SLiC, see Zhao et al. (2023)) and identity preference optimization (IPO, see Azar et al. (2024)), also lie in the scope of offline preference data alignment. On the other hand, methods like online DPO (Dong et al., 2024) and Reinforced Self-Training (ReST, see Gulcehre et al. (2023)) utilize new samples with higher rewards from current model for further policy improvements.

**Offline RL.** Offline RL considers the problem of learning a policy from a fixed datasets where the reward value is provided for each collected transition samples. In (Liu et al., 2019; Nachum et al., 2019), model-free offline RL algorithms are proposed to solve the importance sampling problem. In (Kumar et al., 2020; Cheng et al., 2022), conservatism is incorporated into the value function to avoid overestimation in the offline RL setting. In (Kostrikov et al., 2021), a general algorithm for offline RL is proposed which can avoid any queries to values of out-of-sample actions during training while still enabling multi-step dynamic programming. For the model-based offline RL algorithms, Kidambi et al. (2020) first constructs the estimated world model by utilizing diverse and large transition dataset and then sets hard threshold on the model uncertainty for constructing terminating states to avoid dangerous explorations. In (Yu et al., 2020), the authors proposes a model-based offline policy optimization algorithm (MOPO) which utilizes uncertainty estimation techniques to construct a penalty function to regularize the reward function. Therefore, MOPO can learn a conservative policy which stays in the low-uncertainty region to avoid the distribution shift issue. As a follow-up work, Lu et al. (2021) revisits the design choices of several key hyperparameters in MOPO and fine-tune the corresponding hyperparameters in MOPO to guarantee strong performance. In Yu et al. (2021), the authors propose a model-based offline RL algorithm called COMBO which does not rely on explicit uncertainty estimation. By regularizing the value function on out-of-distribution state-action pairs generated in the estimated world model, COMBO can benefit from the conservatism without requiring explicit uncertainty estimation techniques. As a remark, the algorithms proposed in (Yu et al., 2020; Kidambi et al., 2020; Lu et al., 2021; Yu et al., 2021) all perform conservative policy optimization in a well-constructed dynamics model and the estimated dynamics model keeps fixed during the training of the RL agent. Different from those algorithms mentioned above, (Uehara & Sun, 2021; Rigter et al., 2022) incorporate conservatism into the constructed dynamics model. By adversarially modifying the estimated dynamics model to minimize the value function under the current policy, the proposed methods can learn a robust policy with respect to the environment dynamics and can obtain probably approximately correct (PAC) performance guarantee.

**Hybrid RL.** In Nair et al. (2020), the authors provide a simple framework which can first leverage large offline dataset for pre-training and then quickly perform online fine-tuning of RL policies. In the proposed algorithm, sample-efficient dynamic programming method is used for policy evaluation and a maximum likelihood estimation problem is further designed for solving the optimal policy. In Garg et al. (2022), the authors develop a new framework to solve maximum entropy reinforcement learning, which directly estimates the optimal Bellman operator without replying on explicit access

to the underlying policy. The proposed framework can be used to develop simple but effective reinforcement learning algorithms, which have the flexibility to well in both fully offline or hybrid RL settings. In Ball et al. (2023), the authors first revisit the design choices in existing off-policy methods. Moreover, with a set of minimal but important changes to the existing off-policy RL algorithms, the authors further show that the existing off-policy RL algorithms can achieve reliable performance by leveraging offline data when learning online.

## B   PROOF OF THEOREM 1

In this section, we show the solution to the constrained policy optimization problem defined in (14a)-(14d).

*Proof of Theorem 1.* In this proof, we will first write down the *partial* Lagrangian function, which only considers the constraints (14b)-(14c). After solving the partial Lagrangian function, we will show that the constraint (14d) is satisfied.

Let $\alpha$ and $\zeta := \{\zeta_s | s \in \mathcal{S}\}$ denote the dual variables of the constraints (14b) and (14c), respectively. Then the partial Lagrangian function can be expressed as below:

$$\mathcal{L}(\pi, \alpha, \zeta) := \frac{1}{1-\gamma} \mathbb{E}_{s \sim d_{\pi_{\text{old}}}(\cdot), a \sim \pi(\cdot|s)} \left[ A^{\pi_{\text{old}}}(s, a) \right] + \alpha \left( \epsilon - \mathbb{E}_{s \sim d_{\pi_{\text{old}}}(\cdot)} \left[ D_{\text{KL}} \left( \pi(\cdot|s) || \pi_{\text{old}}(\cdot|s) \right) \right] \right)$$
$$+ \sum_{s \in \mathcal{S}} \zeta_s \left( 1 - \sum_{a \in \mathcal{A}} \pi(a|s) \right).$$

Through taking partial derivative of $\mathcal{L}(\pi, \alpha, \zeta)$ w.r.t. $\pi(a|s)$, we can obtain the following equation:

$$\frac{\partial}{\partial \pi(a|s)} \mathcal{L}(\pi, \alpha, \zeta) = \frac{1}{1-\gamma} d_{\pi_{\text{old}}}(s) A^{\pi_{\text{old}}}(s, a) - \alpha d_{\pi_{\text{old}}}(s) \left( -\log \pi_{\text{old}}(a|s) + \log \pi(a|s) + 1 \right) - \zeta_s.$$

Through setting the partial derivative $\frac{\partial}{\partial \pi(a|s)} \mathcal{L}(\pi, \alpha, \zeta)$ to 0, we obtain

$$\frac{1}{1-\gamma} d_{\pi_{\text{old}}}(s) A^{\pi_{\text{old}}}(s, a) - \alpha d_{\pi_{\text{old}}}(s) \left( -\log \pi_{\text{old}}(a|s) + \log \pi(a|s) + 1 \right) - \zeta_s = 0.$$

Then we obtain the closed-form expression of the optimal policy $\pi^*$ as below:

$$\log \pi^*(a|s) = \frac{A^{\pi_{\text{old}}}(s, a)}{(1-\gamma)\alpha} + \log \pi_{\text{old}}(a|s) - 1 - \frac{\zeta_s}{\alpha d_{\pi_{\text{old}}}(s)}, \tag{27a}$$

$$\pi^*(a|s) = \pi_{\text{old}}(a|s) \exp \left( \frac{A^{\pi_{\text{old}}}(s, a)}{(1-\gamma)\alpha} \right) \exp \left( -1 - \frac{\zeta_s}{\alpha d_{\pi_{\text{old}}}(s)} \right). \tag{27b}$$

Here we can denote $\beta := (1-\gamma)\alpha$. Then according to the expression of $\pi(a|s)$ in (27b), we obtain the following relation:

$$\pi(a|s) \propto \pi_{\text{old}}(a|s) \exp \left( \frac{1}{\beta} A^{\pi_{\text{old}}}(s, a) \right). \tag{28}$$

Based on the constraint (14c), we know that $\pi(\cdot|s)$ is a distribution so that $\sum_{a \in \mathcal{A}} \pi(a|s) = 1$. Therefore, according to the expressions in (27b) and (28), we can obtain the following expression of the optimal policy $\pi^*$ as below:

$$\pi^*(a|s) = \frac{\pi_{\text{old}}(a|s) \exp \left( \frac{1}{\beta} A^{\pi_{\text{old}}}(s, a) \right)}{\sum_{a' \in \mathcal{A}} \pi_{\text{old}}(a'|s) \exp \left( \frac{1}{\beta} A^{\pi_{\text{old}}}(s, a') \right)}.$$

Recall that $A^{\pi_{\text{old}}}(s, a) := Q^{\pi_{\text{old}}}(s, a) - V^{\pi_{\text{old}}}(s)$ has been defined in (10), then we can rewrite the expression of $\pi^*(a|s)$:

$$\pi^*(a|s) = \frac{\pi_{\text{old}}(a|s) \exp \left( \frac{1}{\beta} \left( Q^{\pi_{\text{old}}}(s, a) - V^{\pi_{\text{old}}}(s) \right) \right)}{\sum_{a' \in \mathcal{A}} \pi_{\text{old}}(a'|s) \exp \left( \frac{1}{\beta} \left( Q^{\pi_{\text{old}}}(s, a') - V^{\pi_{\text{old}}}(s) \right) \right)}$$
$$= \frac{\pi_{\text{old}}(a|s) \exp \left( \frac{1}{\beta} Q^{\pi_{\text{old}}}(s, a) \right)}{\sum_{a' \in \mathcal{A}} \pi_{\text{old}}(a'|s) \exp \left( \frac{1}{\beta} Q^{\pi_{\text{old}}}(s, a') \right)}. \tag{29}$$

Then we can define a reference function $W^{\pi_{\text{old}}}(s)$ as below (a trick that was also employed in Garg et al. (2022)):

$$W^{\pi_{\text{old}}}(s) := \beta \log \left( \mathbb{E}_{a \sim \pi_{\text{old}}(\cdot|s)} \left[ \exp \left( \frac{1}{\beta} Q^{\pi_{\text{old}}}(s, a) \right) \right] \right). \tag{30}$$

By plugging the definition of $W^{\pi_{\text{old}}}(s)$ into (9b), we can express the optimal policy $\pi^*(a|s)$ as below:

$$\pi^*(a|s) = \pi_{\text{old}}(a|s) \exp \left( \frac{1}{\beta} \left( Q^{\pi_{\text{old}}}(s, a) - W^{\pi_{\text{old}}}(s) \right) \right). \tag{31}$$

According to the closed-form expression of the optimal policy $\pi^*$ in (31), we obtain that $\pi^*(a|s)$ is non-negative for any state-action pair $(s, a)$ and thus the constraint (14d) is satisfied. $\square$

## C  DETAILS OF THE EXPERIMENT SETTING

We follow the 1b experiment as in (Huang et al. (2024)) and 8b experiment as in (Dong et al. (2024)), where we utilize DeepSpeed ZeRO-3 (Rajbhandari et al. (2020)) and FlashAttention-2 (Dao et al. (2022)) to reduce the memory cost. To accelerate data generation, we use VLLM (Kwon et al., 2023) for inference. We use eight NVIDIA A100-40G to do the training with per device batch size of 64 for 1b model and per device batch size of 16 for 8b model. We train all models with bfloat16 precision. We set the learning rate to be 3e-6 for 1b model and 5e-7 for 8b model with the cosine learning rate scheduler. We consider the max sequence length to be 565 for 1b models and 4096 for 8b models.

We also list the metric and number of shots used for LLM evaluation on each dataset.

| Dataset | Arc Challenge | TruthfulQA MC2 | Winogrande | GSM-8K | HellaSwag | MMLU |
|---------|---------------|----------------|------------|--------|-----------|------|
| Metric | acc_norm | acc | acc | strict-match | acc_norm | acc |
| Num. of Shots | 25 | 0 | 5 | 5 | 10 | 5 |

Table 2: A summarization of the benchmarks we use in this work. We list the metric and number of shots used for LLM evaluation on each dataset.

## D  MORE EXPERIMENT RESULTS

In this section, we provide more experiment results. we include Tables 3 and 4, which correspond to Table 1 in the main body, as well as Table 5, which corresponds to Figure 4 in the main body.

| Tasks Metrics | Arc Challenge acc_norm | TruthfulQA MC2 acc | Winogrande acc | GSM8k strict-match | HellaSwag acc_norm | MMLU | Average |
|---------------|------------------------|---------------------|-----------------|---------------------|---------------------|------|---------|
| LLaMA3-SFT | 62.29% | 53.49% | 78.14% | 72.55% | 81.03% | 64.49% | 68.66% |
| SPR-round1 | 62.54% | 54.09% | 77.51% | 73.77% | 81.14% | 65.08% | 69.02% |
| SPR-round2 | 62.80% | 54.51% | 77.82% | 75.13% | 81.31% | 65.17% | 69.46% |
| SPR-round3 | 63.14% | 55.01% | 78.06% | 75.36% | 81.34% | 65.08% | 69.66% |
| SPR-round4 | 63.48% | 55.54% | 78.37% | 75.81% | 81.49% | 65.25% | 69.99% |
| SPR-round5 | 63.32% | 56.03% | 78.14% | 75.97% | 81.36% | 65.22% | 70.01% |

Table 3: Performance of Policy in Open LLm Leaderboard for for Best-of-16 algorithm.

| Tasks
Metrics | Arc Challenge
acc_norm | TruthfulQA MC2
acc | Winogrande
acc | GSM8k
strict-match | HellaSwag
acc_norm | MMLU | Average |
|---|---|---|---|---|---|---|---|
| LLaMA3-SFT | 62.29% | 53.49% | 78.14% | 72.55% | 81.03% | 64.49% | 68.66% |
| SPR-round1 | 62.46% | 54.27% | 77.66% | 74.30% | 81.21% | 64.94% | 69.14% |
| SPR-round2 | 62.97% | 54.81% | 77.58% | 75.06% | 81.29% | 65.13% | 69.47% |
| SPR-round3 | 63.57% | 55.51% | 78.06% | 75.97% | 81.43% | 65.19% | 69.96% |
| SPR-round4 | 63.91% | 55.86% | 77.82% | 76.04% | 81.54% | 65.35% | 70.09% |
| SPR-round5 | 63.57% | 56.14% | 77.74% | 76.12% | 81.68% | 65.34% | 70.10% |

Table 4: Performance of Policy in Open LLm Leaderboard for Best-of-16 Re-weighting algorithm.

| Tasks
Metrics | Arc Challenge
acc_norm | TruthfulQA MC2
acc | Winogrande
acc | GSM8k
strict-match | HellaSwag
acc_norm | MMLU | Average |
|---|---|---|---|---|---|---|---|
| LLaMA3-SFT | 62.29% | 53.49% | 78.14% | 72.55% | 81.03% | 64.49% | 68.66% |
| SPR-round1 | 62.37% | 53.89% | 77.90% | 74.45% | 81.03% | 65.00% | 69.44% |
| SPR-round2 | 63.14% | 54.74% | 77.82% | 75.21% | 81.20% | 65.17% | 69.88% |
| SPR-round3 | 63.31% | 55.15% | 77.35% | 75.89% | 81.27% | 65.20% | 69.86% |
| SPR-round4 | 63.91% | 55.62% | 78.22% | 77.03% | 81.36% | 65.23% | 70.23% |
| SPR-round5 | 63.82% | 56.07% | 77.82% | 78.01% | 81.51% | 65.33% | 70.43% |
| SPR-round6 | 63.74% | 56.41% | 78.45% | 77.94% | 81.50% | 65.38% | 70.57% |
| SPR-round7 | 63.99% | 56.74% | 78.22% | 77.94% | 81.57% | 65.41% | 70.65% |
| LLaMA3-DPO-iter1 | 63.31% | 57.19% | 78.14% | 74.30% | 80.00% | 64.65% | 69.60% |
| LLaMA3-DPO-iter2 | 65.36% | 60.02% | 77.43% | 70.96% | 81.56% | 63.95% | 69.88% |

Table 5: Performance of Policy in Open LLm Leaderboard for Best-of-32 SPR algorithm.