# OpenReview forum: "Policy optimization can be memory-efficient: LLM Alignment Through Successive Policy Re-weighting (SPR)"
_ICLR.cc/2025/Conference — Submitted to ICLR 2025_

### Official Review · Reviewer_Z1Q9 · 2024-10-27

**Soundness:** 3
**Presentation:** 2
**Contribution:** 2
**Rating:** 3
**Confidence:** 3

**Summary:**

This paper proposes an off-policy alignment method for LLMs based on a well-known weighted supervised learning technique in offline RL. The method offers the advantage of comparable peak memory usage to SFT. The authors provide empirical evidence demonstrating the effectiveness of the proposed SPR method.

**Strengths:**

* The problem studied is significant in the age of LLMs, and the proposed solution aligns well with off-policy RL literature.

* The paper presents empirical evidence supporting the effectiveness of the proposed method.

**Weaknesses:**

While the method is reasonable and useful, the reviewer identified several limitations in the manuscript:

* **Novelty**: The proposed method primarily aligns with existing offline RL literature [1], with its unique technical contribution being the estimation of the normalization factor. However, there is no ablation study assessing the impact of these design choices on performance. Additionally, similar offline RL techniques have been utilized for LLM alignment, such as [2]. A comprehensive comparison with related work in LLM alignment is needed to highlight the true novelty of this study.

* **Contribution**: Although the paper emphasizes the low cost of the proposed method, this has already been achieved by several other studies, e.g., [3]. Furthermore, several works have accomplished training-free alignment [2][4], suggesting that the contribution of this study may not be as significant.

* **Writing**: The overall writing lacks clarity. Sections 2.2 and 3 include large portions of well-known derivations and theorems that could be moved to the appendix to allow for greater emphasis on the study's unique contributions. Additionally, two algorithms are presented, but only the latter one is used. Deleting the first or moving it to the appendix would be beneficial. There is also a typo: $Q_{\phi}$ in Algorithm 2 is not utilized.

* **Experiment Comparison**: (1) The comparison is limited, with several relevant iterative fine-tuning works, such as iterative DPO and Remax [3], and SPIN [5], not included. (2) Prior iterative fine-tuning methods tend to decline after about three iterations. Why does the proposed method sustain performance through seven iterations? Are there any critical design factors?

* **Ablation Study**: The paper lacks a comprehensive ablation study, leaving readers uncertain about the effects of the proposed method's design choices.


**References**

[1] Peng X B, Kumar A, Zhang G, et al. Advantage-weighted regression: Simple and scalable off-policy reinforcement learning[J]. arXiv preprint arXiv:1910.00177, 2019.

[2] Snell C, Kostrikov I, Su Y, et al. Offline rl for natural language generation with implicit language q learning[J]. arXiv preprint arXiv:2206.11871, 2022.

[3] Li Z, Xu T, Zhang Y, et al. Remax: A simple, effective, and efficient reinforcement learning method for aligning large language models[C]//Forty-first International Conference on Machine Learning. 2023.

[4] Mudgal S, Lee J, Ganapathy H, et al. Controlled decoding from language models[J]. arXiv preprint arXiv:2310.17022, 2023.

[5] Chen Z, Deng Y, Yuan H, et al. Self-play fine-tuning converts weak language models to strong language models[J]. arXiv preprint arXiv:2401.01335, 2024.

**Questions:**

Please see the weaknesses part.

---

### Official Review · Reviewer_C5yp · 2024-11-02

**Soundness:** 2
**Presentation:** 2
**Contribution:** 2
**Rating:** 3
**Confidence:** 4

**Summary:**

The paper’s goal is to implement memory efficient LLM alignment. Typical LLM alignment goes through an online RL algorithm like PPO, which requires loading the model, reference model and reward models into memory.

This paper introduces SPR, which is an AWR [11] like method that does iterative optimization, where it only requires to load either the policy or the value method into memory. It uses a weighted supervised learning approach to the subsequent policy alignment, which they show has lower memory requirement compared to DPO/PPO and has comparable performance across a range of benchmark tasks.

**Strengths:**

1. The paper is well-structured and easy to follow.
2. Considering the memory requirement of alignment algorithms can be an important research direction, which has not been explored much until recently. Kudos to the authors for doing this.
3. The paper’s method is interesting and seems to have decent performance.

**Weaknesses:**

**(Missing comparisons)**


Overall, the comparisons in this paper seem incomplete:

1. Since this paper is using a method similar to RWR/AWR, comparing with RWR/AWR [4] would be important.
2. The reward model used for this task seems not super strong. Why not use GPT-4 as a judge, or one of the stronger reward models in 7B-8B parameter range from RewardBench? (https://huggingface.co/spaces/allenai/reward-bench)
3. Where are the other baselines in Figure 3?
4. If the loading of the value model/memory consumption is an issue, and the solution is to sample multiple responses, then a more appropriate comparison instead of PPO is GRPO [7]. The paper is missing this.
5. More datasets. TL;DR is well-known to be not a comprehensive dataset anymore. Experiments on some higher quality dataset, such as Ultrafeedback would make the paper stronger.
6. **The paper says it is beating other baselines, but PPO still seems to be much better for most of the training in Figure 2**.

**(Further concerns)**

The paper mentions DPO requiring a reference policy and hence becoming memory inefficient. There are many ways to mitigate this: there are reference policy-free variants of DPO, for example, SimPO [9]. Alternatively, one can just pre-compute the reference log probs when the reference model + dataset buffer is fixed, see Agent-Q [10]. The main claimed strength for this paper is memory optimization. Therefore not considering these other options/comparing with them in the paper is a big weakness for this paper, since they are also using similar tricks (only loading one model (value or policy) into memory in a single step.


**(Runtime comparisons)**

The paper mentions memory requirement comparisons between multiple algorithms. While that is fine, generally methods that require online sampling from the policy can be quite slow compared to methods that don’t, due to how sampling often is implemented in public codebases. I am quite curious about what is the runtime comparison, between a single run of DPO vs a single run of SPR, on the same dataset and the same hardware? Also, runtime comparison can be slower/faster based on the particular implementation, so a comparison of required FLOPs etc., is also fine. The paper would have been stronger if it discussed this trade-offs better.


**(Missing citations)**

The paper is missing key literature comparing online/offline data and their trade-offs for LLM alignment, such as [1].

**Questions:**

1. Question about PPO

>Given a reward model, RLHF utilizes standard policy optimization algorithm such as PPO at the policy optimization step, whose naive implementation requires loading four models at the same time, namely the reward, the policy, and two value function models

I am looking at the TRL PPO implementation. I am uncertain why one would need two value models? Ideally one can implement PPO with 3 models: reward, policy and a reference policy. Looking at equation (2) of [3], these are 3 models we need. One can additionally use a value model, but I thought it was optional.

2. Question about theorem 1

$\beta$ is defined to be $\frac{1}{(1 - \gamma) \alpha}$, but I could not find what is $\alpha$. Could the authors clarify this? I might have also missed this, apologies in that case.

3. Question about equation 23

What is the motivation for modeling using Gumbell noise? Could the authors explain this in a bit more detail in their response?

4. Question about Figure 2

Why are DPO and PPO horizontal lines in Figure 2? How much data was PPO/DPO trained on? Were the hyper-parameters etc tuned to make them fair comparisons? This is not clear.

5. Question about implementation of best-of-N

How is best-of-N precisely implemented? Is it similar to ReST [5] / SuperHF [6]? The details need to be clarified. Important citations to these papers are also missing.

# References

[1] Preference Fine-Tuning of LLMs Should Leverage Suboptimal, On-Policy Data, https://arxiv.org/abs/2404.14367

[2] TRL PPO implementation: https://github.com/huggingface/trl/blob/main/trl/trainer/ppo_trainer.py

[3] Training language models to follow instructions with human feedback, https://arxiv.org/abs/2203.02155

[4] Aligning Language Models with Offline Learning from Human Feedback, https://arxiv.org/pdf/2404.14367

[5] Reinforced Self-Training (ReST) for Language Modeling, https://arxiv.org/abs/2308.08998

[6] SuperHF: Supervised Iterative Learning from Human Feedback, https://arxiv.org/abs/2310.16763

[7] DeepSeekMath: Pushing the Limits of Mathematical Reasoning in Open Language Models, https://arxiv.org/abs/2402.03300

[8] UltraFeedback: Boosting Language Models with Scaled AI Feedback, https://arxiv.org/abs/2310.01377

[9] SimPO: Simple Preference Optimization with a Reference-Free Reward, https://arxiv.org/abs/2405.14734

[10] Agent Q: Advanced Reasoning and Learning for Autonomous AI Agents, https://arxiv.org/abs/2408.07199

[11] Advantage-Weighted Regression: Simple and Scalable Off-Policy Reinforcement Learning, https://arxiv.org/abs/1910.00177

---

### Official Review · Reviewer_tXJV · 2024-11-03

**Soundness:** 2
**Presentation:** 2
**Contribution:** 2
**Rating:** 5
**Confidence:** 3

**Summary:**

This paper introduces a novel online alignment framework utilizing the TRPO objective to reduce computational and memory requirements. The authors initially derive the relationship between the policy, Q-function, and W-function within the TRPO framework. Subsequently, they introduce a surrogate function to make a better estimation of the W-function.

Experimental results show that the proposed method surpasses DPO in various environments and across different LLMs, while also demonstrating lower memory usage compared to DPO.

**Strengths:**

1. The idea of integrating TRPO into alignment strategies is intriguing.

2. The proposed method outperforms DPO in a range of environments and across various LLM.

**Weaknesses:**

1. I do not fully understand the advantage of using Equation (26) over Equation (23), as Equation (26) introduces an additional reference function $ W_\phi$ that has an extra optimization process. This process consumes more time and memory in pursuit of a better estimation. However, whether the estimator is better depends on whether the authors have developed a quick and precise solver for Equation (26). If the solution is suboptimal, it might perform worse than the empirical estimator from Equation (22). The authors should provide a more detailed discussion on the solver and its superiority over the closed-form solution presented in Equation (23).



2. In Algorithm 1, it is unclear how the minimization problems in Steps 4, 5, and 6 are solved. Moreover, As steps 4, 5, and 6 are executed in sequence, there is a concern that if \( \phi \) in Step 4 is not well-solved, it could affect the learning of \( \varphi \) in Step 5.


3. Some proofs should be discussed in more detail. For example, how to obtain $ \pi^*(a \mid s)$ in lines 905-906 requires further explanation. Specifically, how does the equation $ \sum_{a \in \mathcal{A}} \pi(a \mid s) = 1$ imply that
$\pi^*(a \mid s) = \frac{\pi_{\text{old}}(a \mid s) \exp \left(\frac{1}{\beta} A^{\pi_{\text{old}}}(s, a)\right)}{\sum_{a^{\prime} \in \mathcal{A}} \pi_{\text{old}}\left(a^{\prime} \mid s\right) \exp \left(\frac{1}{\beta} A^{\pi_{\text{old}}}\left(s, a^{\prime}\right)\right)}$?
 Because $\pi^*(a \mid s)$ also satisfies Equation (27b), it seems that the authors should demonstrate that
$\sum_{a^{\prime} \in \mathcal{A}} \pi_{\text{old}}\left(a^{\prime} \mid s\right) \exp \left(\frac{1}{\beta} A^{\pi_{\text{old}}}\left(s, a^{\prime}\right)\right) = \frac{1}{\exp \left(-1 - \frac{\zeta_s}{\alpha d_{\pi_{\text{old}}}(s)}\right)} $ to make the desired result $\pi^*(a \mid s) = \frac{\pi_{\text{old}}(a \mid s) \exp \left(\frac{1}{\beta} A^{\pi_{\text{old}}}(s, a)\right)}{\sum_{a^{\prime} \in \mathcal{A}} \pi_{\text{old}}\left(a^{\prime} \mid s\right) \exp \left(\frac{1}{\beta} A^{\pi_{\text{old}}}\left(s, a^{\prime}\right)\right)}$ established.



4. Regarding the experiments, my primary concern is in Figure 2, where SPR N8 requires five iterations to achieve a performance comparable to DPO, which is only trained in one iteration. This does not seem like a fair comparison. It would be beneficial to see additional training iterations for DPO to ensure a more fair comparison.

Moreover, the authors do not provide standard deviation or statistical significance for Table 5. It would be advantageous to test the proposed method across different random seeds to validate its robustness and repeatability.

**Questions:**

Please address the questions raised above.

Additionally, could the authors provide an analysis of the space complexity for the various methods discussed? This would help readers better understand the theoretical memory reductions by the proposed techniques.

---

### Official Review · Reviewer_AJa2 · 2024-11-04

**Soundness:** 2
**Presentation:** 2
**Contribution:** 2
**Rating:** 3
**Confidence:** 2

**Summary:**

This paper introduces a method enabling training LLMs without the need to explicitly learn a value function. Experimental results indicate that this approach is more memory-efficient, requiring less GPU memory than PPO and DPO. Additionally, the proposed method demonstrates superior performance compared to SFT and DPO on both the TL;DR benchmark and the Open LLM Leaderboard.

**Strengths:**

1. The research problem addressed by the paper is intriguing, particularly in scenarios where GPU memory consumption is a primary concern.
2. The proposed method does not require training a value function, resulting in a about 50% reduction in GPU memory usage compared to PPO.
3. Experimental results demonstrate that the proposed method outperforms SFT and DPO in 3 out of 6 benchmarks.

**Weaknesses:**

1. The paper dedicates three pages (pages 3–5) to preliminaries, most of which could be moved to the appendix, retaining only the essential information in the main text.
2. Equations 16 to 20 appear to be directly sourced from [1], specifically equations 37 to 39. These should also be placed in the preliminaries section.
3. While the design of component W seems to be the paper's primary contribution, there is no ablation study on this design choice. This omission makes it unclear whether the reported improvements are primarily due to the proposed approach in the methods section.

[1] Peng, Xue Bin, et al. "Advantage-weighted regression: Simple and scalable off-policy reinforcement learning." arXiv preprint arXiv:1910.00177 (2019).

**Questions:**

1. What is the number of data points used to train your model in each iteration? Does this align with the amount of data utilized for training SFT and DPO?
2. Other questions have been listed in the weaknesses section. I am open to discussing them further if the authors can clarify my concerns or if I have misunderstood any aspects.

---

### Meta-Review · Area_Chair_3Y3x · 2024-12-19

**Metareview:**

This paper proposes a strategy for reducing memory consumption of LLM alignment via a novel training algorithm. The reviewers raised significant concerns about the novelty compared to existing techniques. In particular, the techniques proposed in the paper appear to be a straightforward adaptation of advantage weighted regression to LLM alignment. There were also concerns about the clarity of the exposition. There was no rebuttal, so the concerns were not addressed. Thus, I believe the paper should be rejected.

**Additional Comments On Reviewer Discussion:**

The authors did not provide a rebuttal, so there was no discussion.

---

### Decision · Program_Chairs · 2025-01-22

Reject